# Al$_2$O$_3$ Ultra-Thin Films Deposited by PEALD for Rubidium Optically Pumped Atomic Magnetometers with On-Chip Photodiode

Florival M. Cunha [1,2,*], Manuel F. Silva [1,2], Nuno M. Gomes [1,2] and José H. Correia [1,2]

1 Center for Microelectromechanical Systems—CMEMS-UMinho, Department of Industrial Electronics, University of Minho, 4800-058 Guimarães, Portugal
2 LABBELS—Associate Laboratory, 4800-122 Braga, Portugal
* Correspondence: florival.cunha@cmems.uminho.pt

**Abstract:** This communication shows the recipe for plasma-enhanced atomic layer deposition (PEALD) Al$_2$O$_3$ ultra-thin films with thicknesses below 40 nm. Al$_2$O$_3$ ultra-thin films were deposited by PEALD to improve the rubidium optically pumped atomic magnetometers' (OPMs) cell lifetime. This requirement is due to the consumption of the alkali metal (rubidium) inside the vapor cells. Moreover, as a silicon wafer was used, an on-chip photodiode was already integrated into the fabrication of the OPM. The ALD parameters were achieved with a GPC close to 1.2 Å/cycle and the ALD window threshold at 250 °C. The PEALD Al$_2$O$_3$ ultra-thin films showed a refractive index of 1.55 at 795 nm (tuned to the D1 transition of rubidium for spin-polarization of the atoms). The EDS chemical elemental analysis showed an atomic percentage of 58.65% for oxygen (O) and 41.35% for aluminum (Al), with a mass percentage of 45.69% for O and 54.31% for Al. A sensitive XPS surface elemental composition confirmed the formation of the PEALD Al$_2$O$_3$ ultra-thin film with an Al 2s peak at 119.2 eV, Al 2p peak at 74.4 eV, and was oxygen rich. The SEM analysis presented a non-uniformity of around 3%. Finally, the rubidium consumption in the coated OPM was monitored. Therefore, PEALD Al$_2$O$_3$ ultra-thin films were deposited while controlling their optical refractive index, crystalline properties, void fraction, surface roughness and thickness uniformity (on OPM volume 1 mm × 1 mm × 0.180 mm cavity etched by RIE), as well as the chemical composition for improving the rubidium OPM lifetime.

**Keywords:** optically pumped atomic magnetometers (OPMs); atomic layer deposition (ALD); ultra-thin films; Al$_2$O$_3$; plasma-enhanced atomic layer deposition (PEALD); trimethylaluminum (TMA)



## 1. Introduction

Optically pumped atomic magnetometers (OPMs) based on alkali vapor cells are currently attractive magnetic sensors due to their relative low cost, low maintenance requirements, and low power consumption [1]. In comparison to other magnetic sensors, the OPMs are robust to temperature, pressure, and vibration, and more attractively offer the possibility of miniaturization through micro-opto-electro-mechanical system (MOEMS) technologies [2,3]. OPMs are the best candidates for the next generation of high-sensitivity magnetic sensors for portable and miniaturized applications in magnetoencephalography and atomic clocks. However, one of the most important problems related to MOEMS-based OPMs is the disappearance of the alkali metal vapor over time. The vapor remains inside the OPM for a time equivalent to the expected lifetime of the alkali vapor [4]. The consumption of the alkali metal inside the vapor cells is attributed to different factors: the reduction of the sodium oxide contained in the glass of the cell, reduction of the silicon dioxide (SiO$_2$) of the glass by the alkali metal, or diffusion of the alkali metal in the glass/silicon envelope [4,5]. Consequently, the improvement of OPMs' cell lifetimes is required by coating the OPM walls. Therefore, the work by S. Woetzel et al. proposed an inner 20 nm-thick alumina

(Al$_2$O$_3$) coating layer for a 2 mm cylindrical cesium OPM cell to improve its lifetime (by a factor of approx. 100) [4]. Additionally, an approximate result was reported by S. Karlen et al. in a 20.25 mm$^2$ square rubidium OPM cell [5].

This work presents the deposition of plasma-enhanced atomic layer deposition (PEALD) Al$_2$O$_3$ ultra-thin films and their optical and chemical characterization in different substrate temperatures, and a comparison to the literature [6–9]. The study of the two main ALD parameters is also addressed: the growth per cycle (GPC) and the ALD window. Moreover, the topography characterization confirms the reduction of roughness and the conformity of the PEALD Al$_2$O$_3$ ultra-thin films for uniformly coating the inside cavity etched by reactive ion etching (RIE). Moreover, the crystalline property, adhesion and surface wettability are addressed and compared with previous works [10–12].

Figure 1 presents the fabricated OPM using silicon and glass wafers and an on-chip photodiode already integrated into the OPM. A 795 nm laser beam, tuned to the D1 transition of rubidium, was used to spin-polarize the atoms, and the intensity of light transmitted through the cell was detected using the integrated photodiode. In a zero magnetic field, the spin magnetic moments aligned with the beam, and transmission of light to the photodiode was maximized. However, a magnetic field perpendicular to the beam caused Larmor precession, rotating the magnetic moments away from alignment. This caused a measurable drop in light transmission [1].

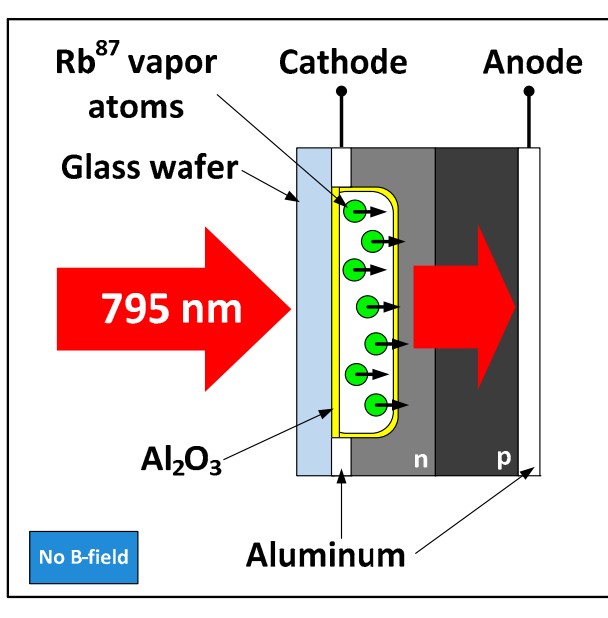 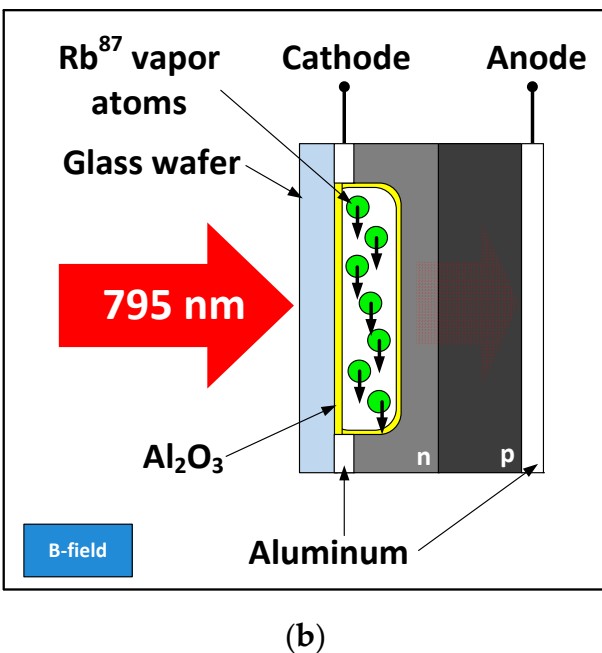

(**a**)  (**b**)

**Figure 1.** Representation of the MOEMS-based OPM: (**a**) without magnetic field; (**b**) with magnetic field.

## 2. Materials and Methods

Al$_2$O$_3$ is an attractive material due to its excellent dielectric properties, good adhesion to many surfaces, and thermal and chemical stability [13–15]. Al$_2$O$_3$ ultra-thin films have many potential applications in diverse areas such as the passivation of electrical contacts, as a protective layer against corrosion by oxygen, and as a gas diffusion layer in polymers, among others [16–27].

Atomic layer deposition (ALD) is a technique capable of overcoming the lack of internal protection of MOEMS-based OPMs with its excellent control over the Al$_2$O$_3$ ultra-thin film thickness (<40 nm). Additionally, ALD allows complex 3D structures (such as the OPM's cell) to be covered with a high-aspect-ratio coating. These characteristics are due to the sequential purging of the precursor and co-reactant with self-limiting binary surface reactions [13,27–29]. The ALD process sequentially grows repeating ultra-thin films in two half-cycles, sub-monolayer by sub-monolayer [13,27,29]. PEALD uses a highly

reactive plasma species of oxygen ($O_2$) or nitrogen ($N_2$) radicals, providing an alternative source of the energy required to grow the ultra-thin films [13,29–31]. PEALD allows a lower temperature (<100 °C) deposition for thermally sensitive substrates, higher-quality films, and an increased growth rate [13,29,31]. Additionally, PEALD uses plasma as a co-reactant, which avoids the oxidation by $H_2O$.

This new OPM design (including a photodiode) had a millimeter-sized cell cavity. A square cavity was fabricated in a 300 μm-thick Si wafer (p-type) with 1 $mm^2$ of area and 180 μm deep by RIE. The RIE masking layer was based in 300 nm-thick aluminum (Al) deposited by the electron beam technique. The Al layer was patterned by photolithography technique using a maskless direct writing laser (DWL) tool and the 2 μm-thick AZ4562 photoresist layer. Finally, an 85% phosphoric acid ($H_3PO_4$) wet etching solution was used to remove the Al layer unexposed by the photoresist. Furthermore, a 2.5 μm/min RIE etching rate was obtained with a controlled 20 mL/min flow of tetrafluoromethane ($CF_4$) as the etching gas and $O_2$-plasma gas flow of 10 mL/min. The RIE power was fixed at 200 W and obtained 35 Pa of reactor pressure.

The deposition of the PEALD $Al_2O_3$ ultra-thin films was performed using a SENTECH Instruments GmbH system, which integrates an ellipsometer for precise real-time ultra-thin film thickness monitoring. The trimethylaluminum (TMA) precursor was selected with the $O_2$-plasma as a co-reactant. Initially, the substrate surface was passivated with the OH* functional groups. In the first half-cycle, TMA was exposed and reacted with OH*, creating $CH_4$ as described in Equation (1).

$$OH^* + Al\,(CH_3)_3 \rightarrow OAl\,(CH_3)_2^* + CH_4 \tag{1}$$

where * shows the species attached to the substrate surface [32,33].

In the second half-cycle, the surface was exposed to a highly reactive $O_2$-plasma. This action forms $CO_2$ and $H_2O$ to be purged, and surface passivation (OH*) was prepared for the next cycle as described in Equation (2).

$$AlCH_3^* + 2O_2 \rightarrow AlOH^* + CO_2 + H_2O \tag{2}$$

The PEALD $Al_2O_3$ ultra-thin films were deposited at different temperatures ranging between 100 °C and 400 °C. The reactor was heated up to 100 °C and the precursor's lines were heated up to 125 °C. The $O_2$-plasma was excited at 13.6 MHz by RF (radio frequency) power of 200 W and flow rate of 150 sccm. The reactor vacuum pressure was throttle-controlled at 20 Pa. Table 1 shows the PEALD $Al_2O_3$ ultra-thin film recipe conditions.

**Table 1.** PEALD recipe for the $Al_2O_3$ ultra-thin films.

| N° of Cycles | TMA | | $O_2$-Plasma | |
|---|---|---|---|---|
| | Pulse Time (ms) | Purge Time (s) | Pulse Time (ms) | Purge Time (s) |
| 300 | 60 | 2 | 5000 | 1 |

The on-chip photodiode was fabricated after the RIE etching with an ionic implantation of type n (using phosphorous) on the p-substrate. The cavity was coated by an inner conformal $Al_2O_3$ ultra-thin film (thickness <40 nm). The film thickness took into account the previous studies of rubidium cell lifetime, with validated values between 15 and 22 nm [4,5] but also the surface uniformity after the RIE etching that required higher thicknesses and a tradeoff with the lower absorption of light in the $Al_2O_3$ film. The final deposited $Al_2O_3$ ultra-thin film thickness was 39 nm. The etched silicon wafer cavity was stacked on the top with a glass wafer (facilitates light penetration) by the anodic bonding technique. A small amount of aqueous $RbN_3$ solution was inserted into the cell to obtain metal alkali vapor [4] by exposition to ultraviolet light. OPM fabrication details are shown in Figure 2.

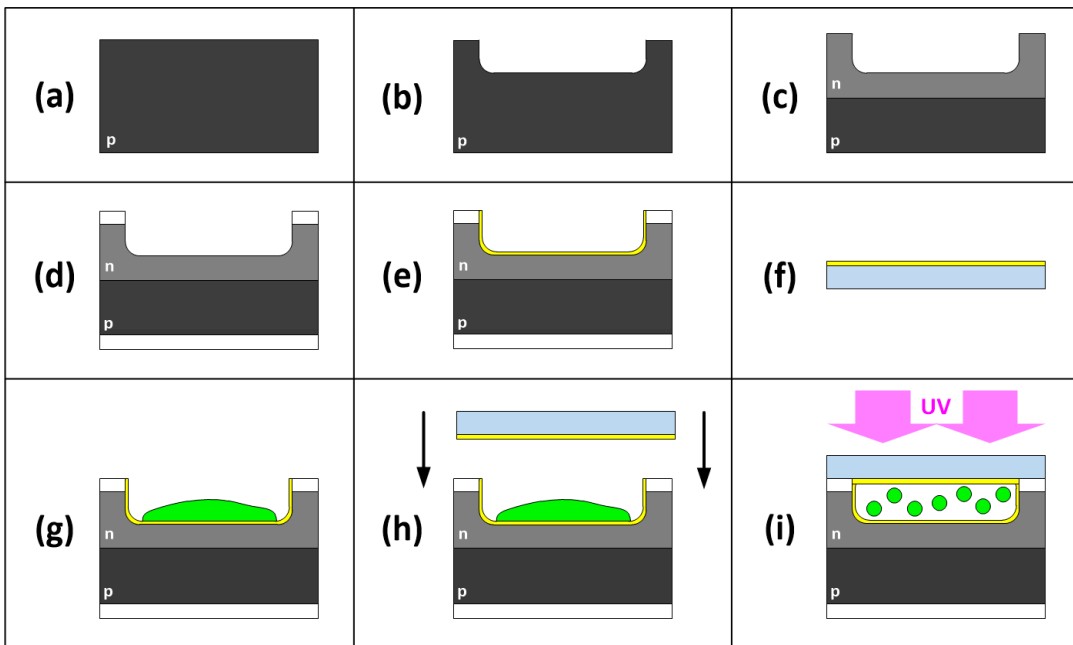

**Figure 2.** Fabrication process overview: (**a**) Si wafer (p-type) of 300 μm thickness as substrate; (**b**) RIE etching of 180 μm-wide through holes in area 1 mm$^2$; (**c**) ionic implantation type n for fabrication of the photodiode; (**d**) aluminum contacts (cathode and anode) for photodiode; (**e,f**) PEALD 39 nm Al$_2$O$_3$ thin film deposition for diffusion barrier; (**g**) aqueous RbN$_3$ solution insertion; (**h**) anodic bonding of 200 μm thick BorofloatR_33 window wafer; (**i**) aqueous RbN$_3$ solution exposition to UV light.

## 3. Results and Discussion

PEALD Al$_2$O$_3$ ultra-thin films were characterized using an in situ ellipsometer. Figure 3 shows the growth of Al$_2$O$_3$ PEALD at 250 °C. A GPC of 1.2 Å/cycle was measured for the growth, as shown in Figure 3a. A more detailed view of the first three cycles is shown in Figure 3b. The time during which the TMA and the O$_2$-plasma were exposed are indicated by the shaped regions. A thickness gain was observed during the TMA exposure, while a decrease in thickness occurred during the O$_2$-plasma exposure.

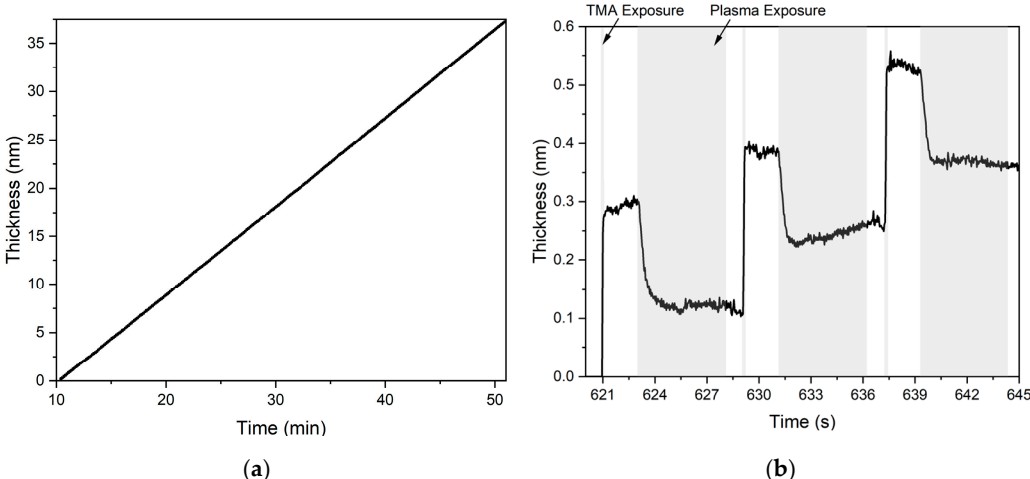

**Figure 3.** Thickness measured in real time for Al$_2$O$_3$ PEALD at 250 °C: (**a**) showing a linear growth over many cycles; (**b**) over the first three cycles, the shaded areas show the time periods during which the TMA and O$_2$-plasma were exposed.

The GPC and the ALD window represent important ALD technique parameters to obtain a reproducible material recipe. Therefore, a series of 300 cycles of PEALD $Al_2O_3$ ultra-thin films were deposited at different temperatures ranging between 100 °C and 400 °C to evaluate the minor variation of the GPC (Figure 4a).

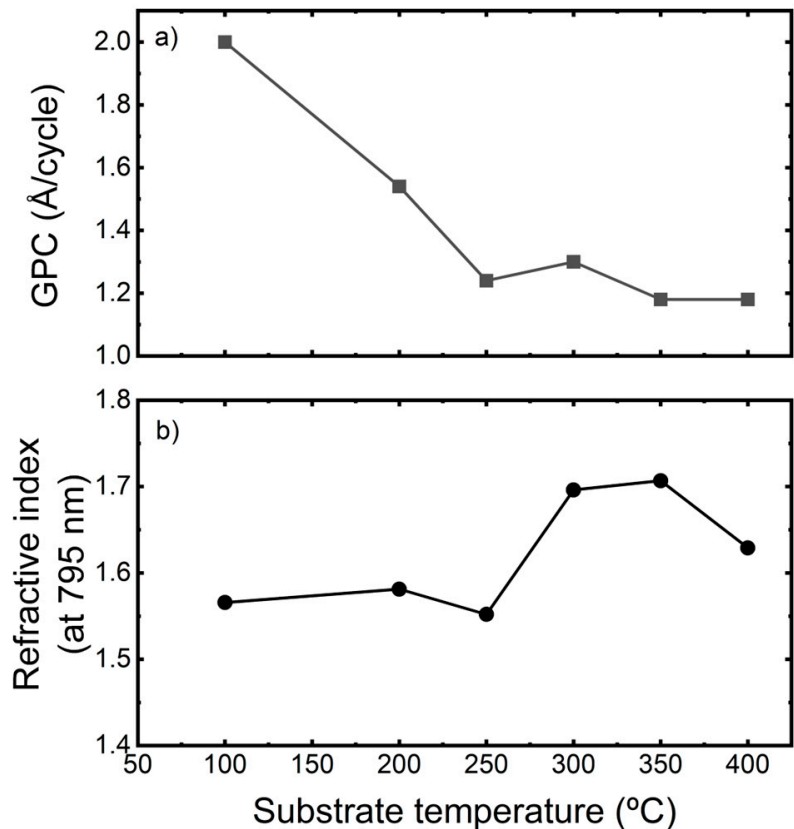

**Figure 4.** (**a**) ALD window; (**b**) refractive index at 795 nm of $Al_2O_3$ ultra-thin films by PEALD at different temperatures.

At substrate temperatures below 250 °C, the GPC was higher than 1.3 Å/cycle; this was related to a condensation phenomenon of the reactants on the surface [13]. Therefore, at 250 °C, the GPC was close to 1.2 Å/cycle and stands for the ALD window threshold.

### 3.1. Optical Characterization

Figure 4b shows the PEALD $Al_2O_3$ ultra-thin film optical characterization performed by a J. A. Woollam Co. (Guimarães, Portugal) spectroscopic ellipsometer model Alpha-SE at 795 nm for different temperatures. The optical data were obtained considering a 2 nm-thick silicon dioxide ($SiO_2$) layer (due to the wafer's contact with the atmosphere) by the CompleteEase (version 6.42, 2018, J. A. Woollam Co., Guimarães, Portugal) software with the Cauchy model and a mean-squared error (MSE) below 3.

For the substrate temperature at 250 °C, the refractive index measured at 795 nm was 1.55. The refractive index increased slightly with temperature, in accordance with earlier publications [6–8]. The measured spectral responsiveness of the on-chip photodiode at 795 nm was 291 mA/W.

### 3.2. Chemical Characterization

The energy dispersive spectroscopy (EDS, Pegasus X4M, EDAX, Guimarães, Portugal) elemental analysis allowed obtaining the correct stoichiometry of the PEALD $Al_2O_3$ ultra-thin films (Figure 5).

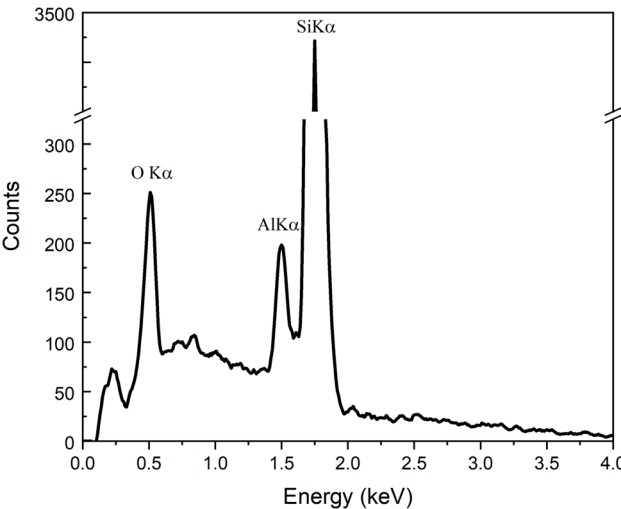

**Figure 5.** EDS elementary analysis of PEALD $Al_2O_3$ ultra-thin film at 250 °C.

The EDS chemical characterization showed a high peak, which represents the substrate Si p-type (100 oriented) and is more relevant because the deposited ultra-thin films thickness was lower than 40 nm. Even so, the EDS of the PEALD $Al_2O_3$ ultra-thin films showed an atomic percentage of 58.65% for oxygen (O) and 41.35% for aluminum (Al) with a mass percentage of 45.69% for O and 54.31% for Al.

This chemical characterization is close to the stoichiometry values for the atomic percentage (60% for O and 40% for Al) and mass percentage (47% for O and 53% for Al).

The X-ray photoelectron spectroscopy (XPS, SCALAB250Xi, ThermoFisher Scientific, Braga, Portugal) technique is a more sensitive surface elemental composition analysis for ultra-thin films below 40 nm thick (Figure 6).

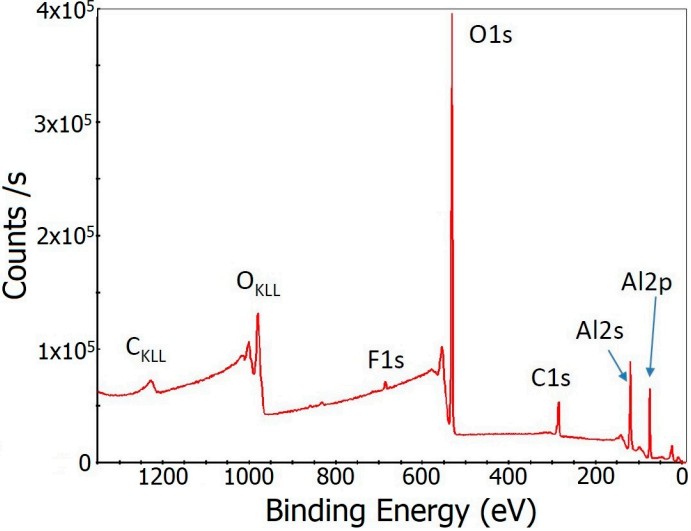

**Figure 6.** XPS spectra for PEALD $Al_2O_3$ ultra-thin film at 250 °C.

The stoichiometric ratio of aluminum to oxygen (Al: O) was found to be close to 34:55 by calculating the XPS peak areas of the Al 2p and O 1s. This value is slightly lower than the expected value of 2:3 which shows that the $Al_2O_3$ ultra-thin film was oxygen rich. The binding energy of 72.8 eV was unobserved (corresponding to the Al–Al bond). This suggests that the deposited $Al_2O_3$ was not oxygen deficient. Additionally, all binding energies were referenced to the C 1s core peak at 284.8 eV. Therefore, the appearance of Al 2s at 119.2 eV and Al 2p at 74.4 eV confirmed the formation of $Al_2O_3$.

### 3.3. Topographic Characterization

Figure 7 displays the scanning electron microscopy (SEM, FEI Nova 200, NanoSEM, Guimarães, Portugal) analysis of a 180 μm-deep cavity coated with a 39 nm-thick PEALD $Al_2O_3$ ultra-thin film to confirm the inner OPM coating layer conformity.

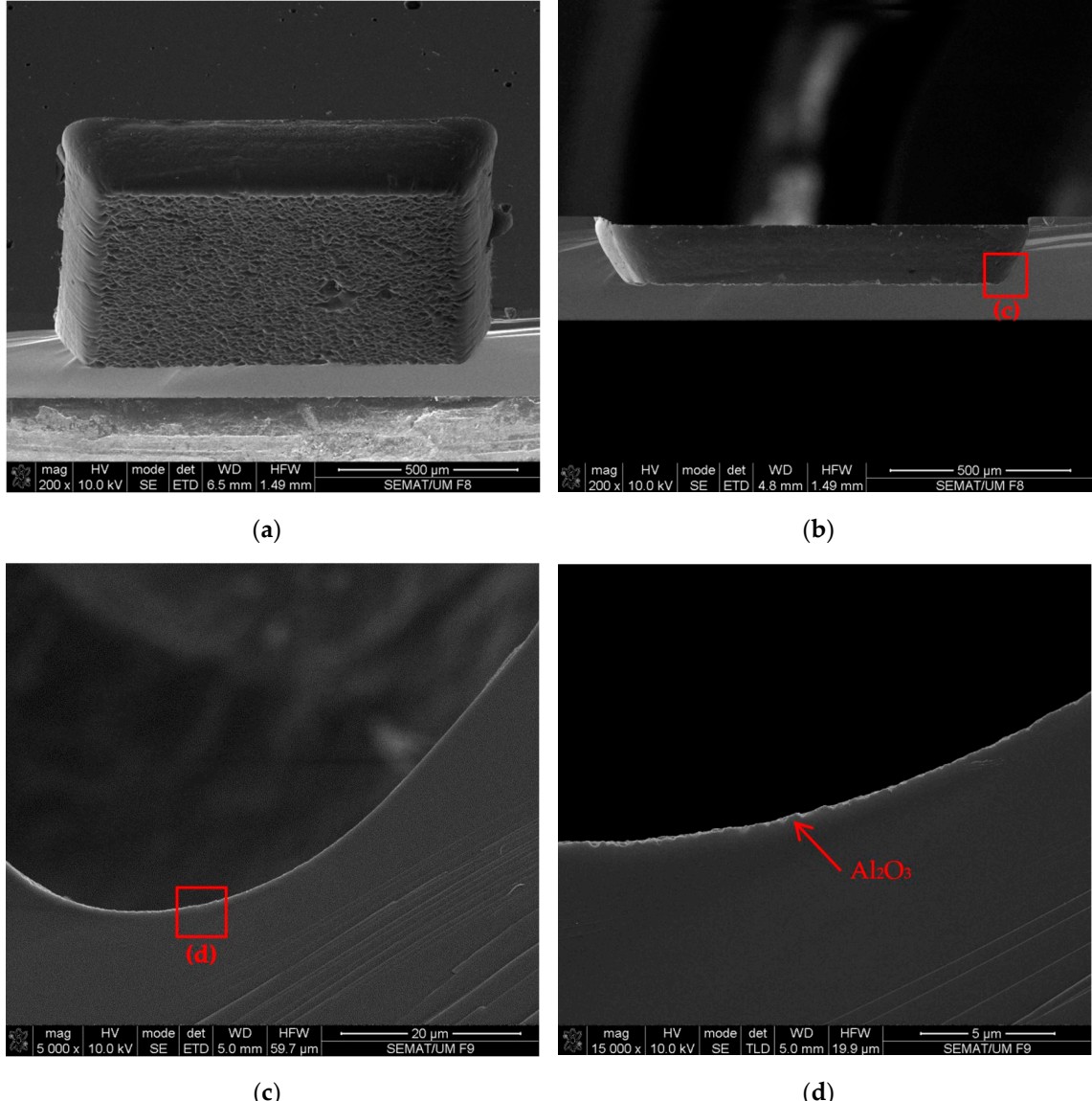

**Figure 7.** SEM analysis: (**a**) 20-degree angled horizontal-cut view of a squared cavity with 1 $mm^2$ of area; (**b**) horizontal-cut view of a 180 μm-deep cavity in a 300 μm-thick Si wafer; (**c**) PEALD $Al_2O_3$ ultra-thin film conformity in the cavity wall; (**d**) detail of the 39 nm PEALD $Al_2O_3$ ultra-thin film (white surface coating).

The SEM images confirmed that the PEALD $Al_2O_3$ ultra-thin film OPM cavity inner coating layer conformity was excellent (at bottom and walls) and the non-uniformity was calculated around 3%.

Profilometer analysis was performed on the PEALD $Al_2O_3$ ultra-thin film and the root-mean-square surface roughness after the RIE etching (180 μm) was approximately 310 nm.

### 3.4. Crystalline Property, Adhesion and Surface Wettability

The XRD measurements suggested that the $Al_2O_3$ films deposited at 250 °C were amorphous, which is in agreement with the reported results by J. N. Ding et al. [10] and Jakschik et al. [11]. Both studied the crystallization behavior of ALD $Al_2O_3$ and found that it was amorphous when the deposition temperature was lower than 900 °C.

Despite the very low thickness (39 nm), the $Al_2O_3$ thin films demonstrated excellent adhesion to the etched substrate (see Figure 7d). The homogeneous microstructure of the $Al_2O_3$ thin films was observed with the use of SEM without any visible delamination. Additionally, a few micro-scratch tests were performed for validating the $Al_2O_3$ coverage after the RIE etching, and the obtained results were in agreement with the literature [10] for silicon substrates.

The surface wettability of the deposited $Al_2O_3$ films was not an issue in this application as the permeability for water vapor of the thin film deposited at low temperatures, in this case at 250 °C, was low, in agreement with [12].

### 3.5. Measurement of the Rubidium Consumption

A quantitative lifetime estimation of the OPM wall coated with 39 nm $Al_2O_3$ and without the $Al_2O_3$ coating is shown in Figure 8; both OPMs were heated at 150 °C. An imaging analysis technique using a microscope, presented in [5], was used to quantify the amount of metallic rubidium present in the OPM and to monitor its consumption over time based on an initial rubidium quantity. Rubidium vapor atoms were created by the UV irradiation of the aqueous $RbN_3$ solution (a three-day exposure) and the surface top window of the OPM was almost covered by metallic rubidium atoms (the start in Figure 8). After this initial condition, the metallic rubidium consumption could be observed over a few days as the surface area covered diminished. A linear consumption was observed in our measurements for the OPM with the $Al_2O_3$ coating.

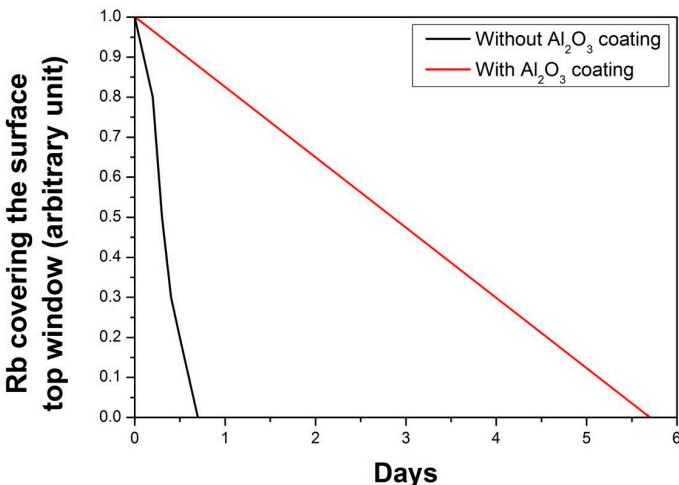

**Figure 8.** Evolution of the amount of rubidium for two OPMs with and without $Al_2O_3$ coating (heated at 150 °C). The measurements were based on the observation of the presence of metallic rubidium on the surface top window of the OPM.

### 4. Conclusions

This paper shows the recipe for 39 nm PEALD $Al_2O_3$ ultra-thin film thickness to improve the rubidium optically pumped atomic magnetometers' (OPMs) lifetimes. An $Al_2O_3$ coating with the thickness of 39 nm showed an enhancement of the resistance and improvement of the surface uniformity after the RIE etching of the silicon substrate with lower absorption of the light in the $Al_2O_3$ film for measurement in the integrated photodiode. The applied coating was fabricated by PEALD, which enables a conformal coverage of the OPM surfaces, including vertical surfaces.

PEALD $Al_2O_3$ ultra-thin film was deposited by controlling their low optical refractive index (lower absorption of the light) in comparison to the literature [6], crystalline properties and thickness uniformity (OPM volume 1 mm $\times$ 1 mm $\times$ 0.180 mm cavity etched by RIE).

The ALD parameters were a GPC close to 1.2 Å/cycle and an ALD window threshold around 250 °C. For depositions at 250 °C, the $Al_2O_3$ refractive index was 1.55 at 795 nm. The characterization of the PEALD $Al_2O_3$ ultra-thin films was realized by EDS chemical elemental analysis and XPS surface elemental composition.

The comparison in [4] is more qualitative as they used cesium vapor cells instead of rubidium vapor cells, but we also conclude that an ultra-thin layer of $Al_2O_3$ can improve the resistance of micro-fabricated atomic vapor cells. As our OPM volume was 0.18 mm$^3$ and in [5] the OPM volume was 3.14 mm$^3$, the OPM lifetime in our case was lower than in [5], but a linear rubidium consumption was observed in both cases. Additionally, the OPM without the $Al_2O_3$ coating showed an abrupt rubidium consumption in both cases.

We can therefore conclude that the UV decomposition of aqueous $RbN_3$ solution together with PEALD $Al_2O_3$ wall coating and an on-chip photodiode is a combination of technologies that enables the low-cost, wafer-level fabrication of MEMS atomic vapor cells with an extended lifetime.

**Author Contributions:** The work presented in this paper was a collaboration of all authors. F.M.C. and M.F.S. made the literature analysis and wrote the first draft of the manuscript. N.M.G. worked at the thin film depositions. J.H.C. corrected, revised, and supervised the manuscript. All authors have read and agreed to the published version of the manuscript.

**Funding:** This work was supported by project MME reference 105399; CMEMS-UMinho Strategic Project UIDB/04436/2020 and UIDP/04436/2020; Infrastructures Micro&NanoFabs@PT, NORTE-01-0145-FEDER-022090, Portugal 2020; and MPhotonBiopsy, PTDC/FIS-OTI/1259/2020.

**Institutional Review Board Statement:** Not applicable.

**Informed Consent Statement:** Not applicable.

**Data Availability Statement:** Data sharing is not applicable to this article.

**Conflicts of Interest:** The authors declare no conflict of interest.

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
