# Peer review of "Al2O3 Ultra-Thin Films Deposited by PEALD for Rubidium Optically Pumped Atomic Magnetometers with On-Chip Photodiode"

_coatings, doi:10.3390/coatings13030638_

Round 1

Reviewer 1 Report

The article is on a very good level.
I have got only a few remarks.
In the article itself, the word used is "Article", but on the website, it is "Communication".

Regarding Figures No. 1, 3, and 7, make sure they do not exceed the page margin on the left.
When it comes to Figure No. 6, a sharper axis description would be more suitable.

Author Response

Dear Reviewer 1,

First, we would like to thanks the reviewer for the comments and proposed suggestions. This is a Communication, it was corrected in the text.

We checked the dimensions of Figures 1,3 and 7.

A sharper axis description in Figure 6 was done.

Also, it was improved the English.

Thank you very much.

Reviewer 2 Report

This paper shows the recipe of 39 nm PEALD Al2O3 ultra-thin film thickness to im- 220 prove the rubidium optically pumped atomic magnetometers (OPMs) lifetime. The paper is well organized. Please revise the conclusion part to breifly summary the main method and progress in this work before publication .

Author Response

Dear Reviewer 2,

First, we would like to thanks the reviewer for the comments and proposed suggestions. The conclusion part was revised and shorten. Also, it was improved the English.

Thank you.

Reviewer 3 Report

This is a contribution to the fabrication of  more  efficient and durable optically pumped atomic manometers by coating the inner side of their silicon cells with alumina. The authors argue that the  plasma enhanced atomic layer deposition technique allows one to achieve a desired uniformity and  reduced roughness of the coating layers.  The deposition of the Al2O3 films  of controllable thickness was performed with the use of a commercial  SENTECH Instruments GmbH system with an integrated ellipsometer for a precise real  time ultra-thin film thickness monitoring. A very instructive Fig. 2 allows the reader to follow the consecutive stages of the manufacturing  process, i.e. the title “recipe”.

The resulting thin layers have been characterised with ellipsometry, energy dispersive, spectroscopy,  X-ray photoelectron spectroscopy and images of scanning electron  microscopy. The surfaces coated according to the recipe turn out about six times more  efficient in the retention of  rubidium atoms essential for the life time of the magnetometer (see Fig. 8). The study is well designed and the variety of characterization methods gives it a value of completeness, although a tip microscopy (AFM, STM) would be also interesting. I suggest its publication in “Coatings”.  

Author Response

Dear Reviewer 3,

First, we would like to thanks the reviewer for the comments and proposed suggestions. The conclusion part was revised and shorten. Also, it was improved the English.

Thank you

Reviewer 4 Report

This paper describes a topic on the recipe of Plasma Enhanced Atomic Layer Deposition (PEALD) Al2O3 ultra-thin films for thickness below 40 nm. Al2O3 ultra-thin films were deposited by PEALD to improve the rubidium optically pumped atomic magnetometers (OPMs) cell lifetime. This requirement is due to the consumption of the alkali metal (rubidium) inside the vapor cells. A silicon wafer is used, an on-chip photodiode is already integrated in the fabrication of the OPM. The ALD parameters were achieved with GPC close to 1.2 Å/cycle and the ALD window threshold at 250 °C. The PEALD Al2O3 ultra-thin films show a refractive index of 1.55 at 795 nm (tuned to the D1 transition of rubidium, for spin-polarization of the atoms). The SEM analysis presents a non-uniformity of around 3%. Finally, the rubidium consumption in the coated OPM was monitored. Therefore, PEALD Al2O3 ultra-thin films were deposited controlling their optical refractive index, crystalline properties, void fraction, surface roughness and thickness uniformity (on OPM volume 1 mm × 1 mm × 0.180 mm cavity etched by RIE), and chemical composition for improving the rubidium OPM lifetime.

The following comments are as below:

1.      It is strongly recommended that the authors should mention clearly the newly developed and /or found point of in section introduction, compared with papers already reported in this field, adding the references is recommended.

2.      How about the crystalline property by using XRD for Al2O3 ultra-thin films?

3.      How about the surface roughness of Al2O3 ultra-thin films?

4.      How about the adhesion of Al2O3 ultra-thin films?

5.      How about the surface wettability of Al2O3 ultra-thin films?

6.      The authors should consider surface roughness of the Al2O3 ultra-thin films because the roughness affects the device performance, such as the gate leakage current and post-processes.

7. Addition of some discussion compared with above comments is recommended.

Author Response

Dear Reviewer 4,

Point 1. It is strongly recommended that the authors should mention clearly the newly developed and /or found point of in section introduction, compared with papers already reported in this field, adding the references is recommended.

Response 1: First, we would like to thanks the reviewer for the comments and proposed suggestions.

The introduction section was modified and added new references.

Point 2. How about the crystalline property by using XRD for Al2O3 ultra-thin films?

Response 2: The XRD measurements suggest that the Al2O3 films deposited at 250 °C are amorphous, which are in agreement with the reported results by J. N. Ding et al. [10] and Jakschik et al. [11]. Both studied the crystallization behavior of ALD Al2O3 and found that it is amorphous when the deposition temperature is lower than 900 °C.

Point 3. How about the surface roughness of Al2O3 ultra-thin films?

Response 3: The absence of Atomic Force Microscopy (AFM) analysis for characterizing the surface roughness of the Al2O3 thin-film inside of the cavity is due the AFM probe dimensions, it cannot enter in the cavity. Therefore, the profilometer analysis was done and the root mean square surface roughness after the RIE etching (180 um) is approximately 310 nm.

Point 4. How about the adhesion of Al2O3 ultra-thin films? 

Response 4: Despite the very low thickness (39 nm) the Al2O3 thin-films demonstrated an excellent adhesion to the etched substrate, see Figure 7d). The homogeneous microstructure of the Al2O3 thin-films was observed with the use of the SEM, without any visible delamination. Also, few micro-scratch tests were done for validating the Al2O3 coverage after the RIE etching, the obtained results are in agreement with the results of J. N. Ding et al. [10] for silicon substrates.

Point 5. How about the surface wettability of Al2O3 ultra-thin films?

Response 5: The surface wettability of the Al2O3 deposited films is not an issue in this application as the permeability for water vapor of the thin film deposited at low temperatures, in this case at 250 °C, is low, H. Li et al. [12] achieved 1.5 × 10−4 g/(m2·day) under ambient conditions of 25 °C and 60% relative humidity. However, due to the large value for surface roughness the trend to increase surface hydrophobicity is present as predicted by the Wenzel's mode and shown by Li-Chun Wang (doi:10.1016/j.surfcoat.2016.08.023).

Point 6. The authors should consider surface roughness of the Al2O3 ultra-thin films because the roughness affects the device performance, such as the gate leakage current and post-processes.

Response 6: The surface roughness of the Al2O3 ultra-thin films affects the device performance, but in this case as the application is a coated closed cell the loss light is minimized. However, the surface roughness can be improved using an Al2O3 thicker layer than 39 nm but increasing the light absorption.

Point 7. Addition of some discussion compared with above comments is recommended.

Response 7: The discussion about the above topics was added to the communication.

Also, it was improved the English.

Thank you.

Round 2

Reviewer 4 Report

The revised manuscript can be accepted.